# scFseCluster: a feature selection-enhanced clustering for single-cell RNA-seq data

Zongqin Wang[1], Xiaojun Xie[1,2], Shouyang Liu[3], Zhiwei Ji[1,2]

**Single-cell RNA sequencing (scRNA-seq) enables researchers to reveal previously unknown cell heterogeneity and functional diversity, which is impossible with bulk RNA sequencing. Clustering approaches are widely used for analyzing scRNA-seq data and identifying cell types and states. In the past few years, various advanced computational strategies emerged. However, the low generalization and high computational cost are the main bottlenecks of existing methods. In this study, we established a novel computational framework, scFseCluster, for scRNA-seq clustering analysis. scFseCluster incorporates a metaheuristic algorithm (Feature Selection based on Quantum Squirrel Search Algorithm) to extract the optimal gene set, which largely guarantees the performance of cell clustering. We conducted simulation experiments in several aspects to verify the performance of the proposed approach. scFseCluster performed very well on eight benchmark scRNA-seq datasets because of the optimal gene sets obtained using the Feature Selection based on Quantum Squirrel Search Algorithm. The comparative study demonstrated the significant advantages of scFseCluster over seven State-of-the-Art algorithms. In addition, our analysis shows that feature selection on high-variable genes can significantly improve clustering performance. In conclusion, our study demonstrates that scFseCluster is a highly versatile tool for enhancing scRNA-seq data clustering analysis.**

## Introduction

Single-cell RNA-sequencing (scRNA-seq) technologies have revolutionized biological research (1, 2). Traditional RNA sequencing (RNA-seq) methods are usually applied to a population of cells, which generates an average gene expression profile that may not capture the heterogeneity of individual cells. In contrast, scRNA-seq provides a higher resolution view of gene expression within individual cells, allowing for the identification and characterization of previously unknown cell types and subtypes and enabling

the detection of rare or low-abundance cell populations. This contributes to a better understanding of cellular transcriptome regulation and the hierarchy of variation (3, 4, 5). Particularly, scRNA-seq can help us understand cell–cell communications (6, 7, 8). Therefore, it provides a new way to explore physiological processes and pathological mechanisms of diseases at the single-cell level and identify new diagnostic markers or new therapeutic targets (9, 10, 11).

scRNA-seq clustering is a critical step for cell-type identification, helping to uncover hidden patterns and potential confounding factors (12, 13). However, the main challenges of clustering scRNA-seq data include the "curse" of dimensionality and the computationally intensive nature of geodesic computations in high-dimensional spaces (14, 15, 16, 17). In addition, sparsity and noise in the data can affect the performance of algorithms (18). Moreover, the heterogeneity of sample composition based on the scRNA-seq data reduces the generalization of a clustering strategy, which makes it difficult to compare the performance of different algorithms (19, 20). To address the above challenges, developing advanced computational approaches for scRNA-seq clustering analysis is urgently needed.

Over the past few years, several computational methods for scRNA-seq clustering have been developed. These methods can be classified into two categories: (1) machine learning (ML)-based methods (21, 22, 23) and (2) deep learning (DL)-based methods (24, 25, 26, 27). In general, ML-based approaches tend to achieve clustering analysis through two independent steps: (1) feature extraction on the original gene expression matrix and (2) clustering the reduced data with traditional models. For example, Satija and coworkers developed Seurat to implement cell clustering by using the Louvain algorithm on the principal components of HVG (high variable gene) matrix (28). Kiselev and colleagues proposed a consensus clustering method SC3 (29). SC3 consists of three steps, including calculating the distances between cells for HVGs, transforming the distance matrices using PCA or graph Laplacian, and then performing K-means clustering to generate a consensus matrix to obtain stable cell clustering. However, ML-based approaches have the following two major limitations: (1) the feature extraction strategies (e.g., PCA) used by the above ML-based

[1]College of Artificial Intelligence, Nanjing Agricultural University, Nanjing, China  [2]Center for Data Science and Intelligent Computing, Nanjing Agricultural University, Nanjing, China  [3]Academy for Advanced Interdisciplinary Studies, Nanjing Agricultural University, Nanjing, China

Correspondence: Zhiwei.Ji@njau.edu.cn

methods are hard to capture nonlinear structures hidden in the scRNA-seq data (30); (2) information loss caused by dimensionality reduction also leads to low accuracy (31, 32). In the past few years, DL-based approaches have been developed for scRNA-seq clustering, including DESC (33), DCA (34), and scDeepCluster (35), etc. Among these representative methods, AutoEncoder (AE) (36, 37) has become the most representative module, which implements the denoising and latent representation of the expression matrix. To some extent, the success of deep clustering strategies depends on extracting the inherent structures of datasets. Particularly, only several latent variables are needed to facility cell clustering and visualization (35, 38, 39). Most recently, several AE-based hybrid models emerged (e.g., scGNN (18), scCAEs (26), scDHA (40)), which appear to provide higher clustering performance. The major limitations of these representative approaches are their high computational cost and low generalization (41). Furthermore, the importance of different genes for scRNA-seq clustering is often overlooked (42, 43). Therefore, it is an attractive strategy to develop an efficient and reliable feature selection framework for scRNA-seq clustering analysis.

In this study, we proposed a novel computational framework scFseCluster for scRNA-seq clustering analysis. With the support of Feature Selection based on Quantum Squirrel Search Algorithm (FSQSSA), a metaheuristics module for feature selection, scFseCluster implemented the clustering task on optimal gene sets and achieved excellent performance on a batch of benchmark datasets with different scales. Further analysis revealed that our proposed method is significantly superior to seven State-of-the-Art (SOTA) algorithms. In summary, scFseCluster is a promising model for accurately identifying cell types from the scRNA-seq data.

# Results

### FSQSSA provides the best performance for gene selection

To test the performance for gene selection, we applied the proposed FSQSSA on eight benchmarking scRNA-seq datasets (Table 1), including Xin (44), Goolam (45), PBMC (46 Preprint), Romanov (47), Darmanis (48), Usoskin (49), Monroto (50), and Hrvatin (51). These datasets were generated from five representative sequencing platforms (3). Then, we compared the performance of FSQSSA

method with four metaheuristic algorithms, including Squirrel Search Algorithm (Squirrel) (52), Enhanced Salp Swarm Algorithm (Salp) (53), Artificial Bee Colony (ABC) (54), and Genetic Algorithm (GA) (55). As shown in Fig 1A, we find that the optimal gene sets obtained from FSQSSA are obviously before other SOTA algorithms in the fitness. FSQSSA has the strongest convergence performance among these metaheuristic algorithms (Fig 1B). In particular, FSQSSA exhibited extremely fast convergence rate on datasets Xin, Goolam, and Romanov. Our results indicate that FSQSSA performs excellent in gene selection of scRNA-seq data (Supplemental Data 1 and Supplemental Data 2).

### The robustness of FSQSSA

To analyze the robustness of FSQSSA, we further evaluated its performance from the following two aspects. First, we checked if the convergence performance of FSQSSA was steady across multiple replicates. From Fig 2, we can see that the convergence curves of FSQSSA vary very little on seven datasets except for Romanov, indicating that population initialization has no significant impact on the convergence performance. Also, the sample size did not significantly affect the convergence of FSQSSA. We then evaluated if the inherent randomness of FSQSSA affects FSQSSA's output. As shown in Fig S1, our method provides reliable and optimal feature subsets in parallel computing. Overall, the FSQSSA has strong robustness.

### scFseCluster outperforms the existing methods

To investigate the clustering performance, we applied scFseCluster to eight scRNA-seq datasets with known cell types to analyze its clustering performance. A total of six metrics (ARI, RI, AMI, NMI, ACC, and FMI) were used (53). As shown in Fig 3, scFseCluster exhibits excellent performance on all of the datasets, especially datasets Xin, Goolam, Usoskin, and Hrvatin (Supplemental Data 3).

We also compared the performance of scFseCluster with seven existing methods, including Seurat (28), scDeepCluster (35), CIDR (56), DESC (33), SINCERA (13), SC3 (29), and SIMLR (57). Our results show that scFseCluster outperforms all other methods. It is worth mentioning that scFseCluster exhibits best on the dataset Goolam, with all metrics close to 1 (Fig 3B). Seruat and SC3 achieved the same average performance on this dataset, ranking #2 with a mean value

**Table 1. A summary of real single-cell RNA sequencing used in our experiment.**

| Dataset | Accession ID | Tissue | Classes | Cells | Genes | PMID |
|---------|--------------|--------|---------|-------|-------|------|
| Goolam | E-MTAB-3321 | Mouse embryo | 5 | 124 | 41,428 | 27015307 |
| Darmanis | GSE59739 | Human brain | 9 | 328 | 22,085 | 26060301 |
| Usoskin | GSE59739 | Mouse brain | 4 | 622 | 25,334 | 25420068 |
| Xin | GSE81608 | Human pancreas | 4 | 1,600 | 39,851 | 27667665 |
| Romanov | GSE74672 | Mouse brain | 7 | 2,881 | 24,341 | 27991900 |
| 10X PBMC | 10X Genomics | Human PBMC | 8 | 4,271 | 16,653 | — |
| Montoro | GSE103354 | Human pancreas | 7 | 7,193 | 27,716 | 30069044 |
| Hrvatin | GSE102827 | Mouse visual cortex | 8 | 48,266 | 25,187 | 29230054 |

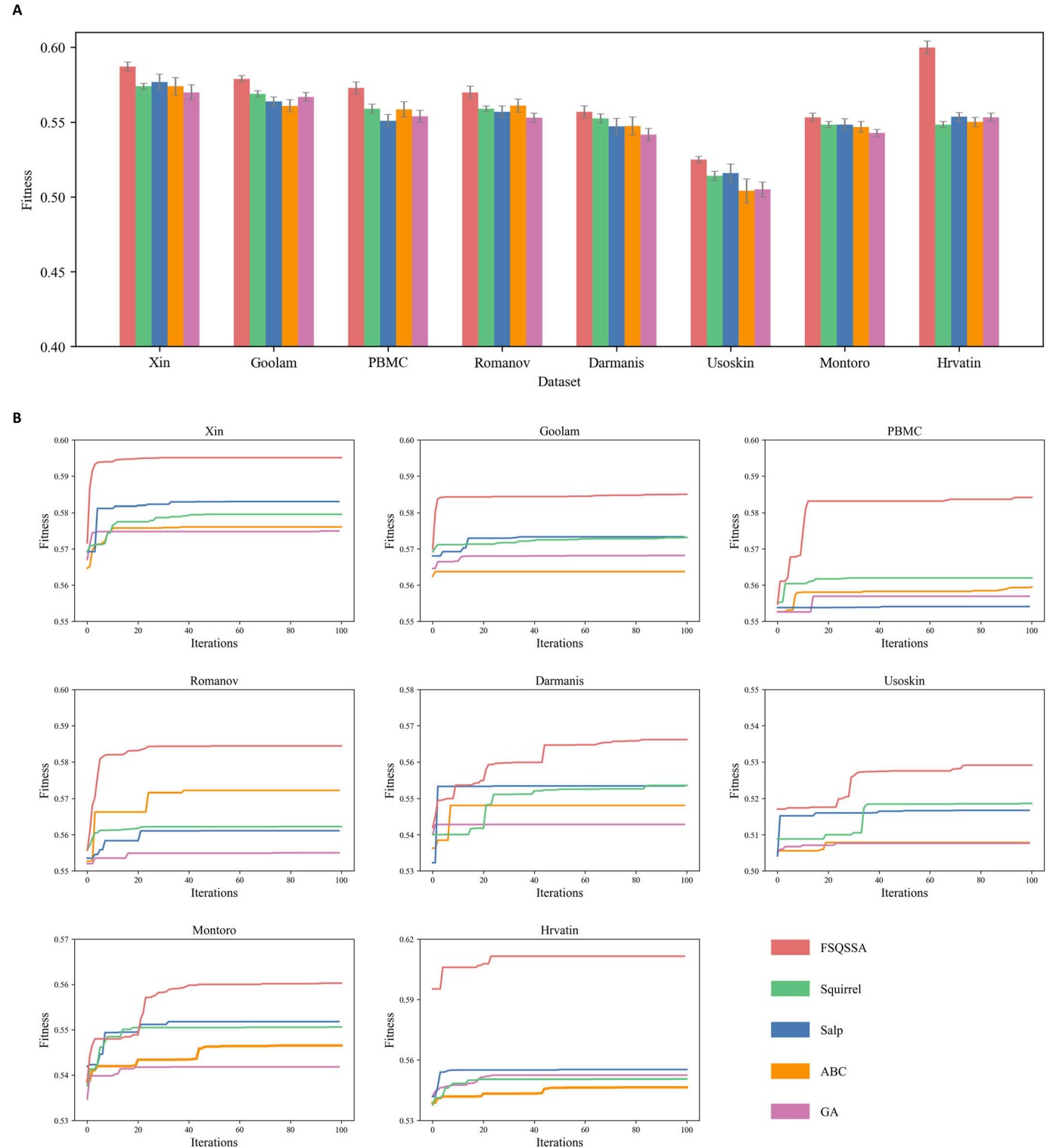

**Figure 1. Comparison analysis between Feature Selection based on Quantum Squirrel Search Algorithm and other four metaheuristics algorithms.**
**(A)** The fitness value of the optimal feature subset for each dataset. **(B)** The convergence curve for all five algorithms.

of 0.731. Similarly, scFseCluster is significantly ahead of other methods on the dataset Xin, Romanov, Darmanis, Usoskin, and Hrvatin (Fig 3A, D–F, and H). We also noticed that SC3 and SINCERA

performed well on very small datasets, whereas scDeepCluster and DESC worked well on datasets with large sample sizes. Furthermore, the comparison analysis on the PBMC dataset, a gold-standard

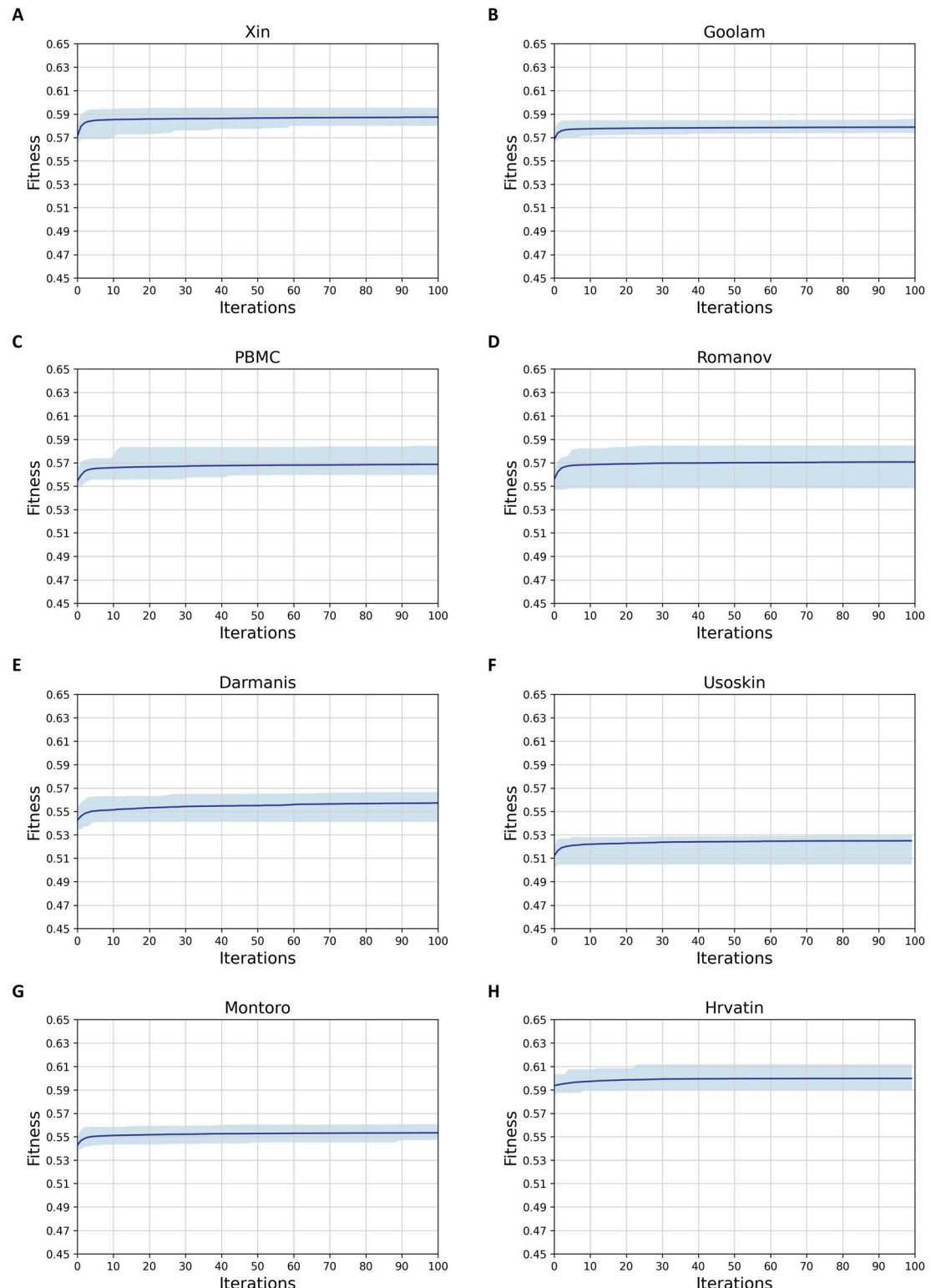

**Figure 2. Robustness analysis of the Feature Selection based on Quantum Squirrel Search Algorithm algorithm on eight single-cell RNA sequencing datasets.**
The light blue area represents the fitness floating interval, and the dark blue line represents the mean fitness values as that number of iterations. **(A)** Xin; **(B)** Goolam; **(C)** PBMC; **(D)** Romanov; **(E)** Darmanis; **(F)** Usoskin; **(G)** Montoro; and **(H)** Hrvatin.

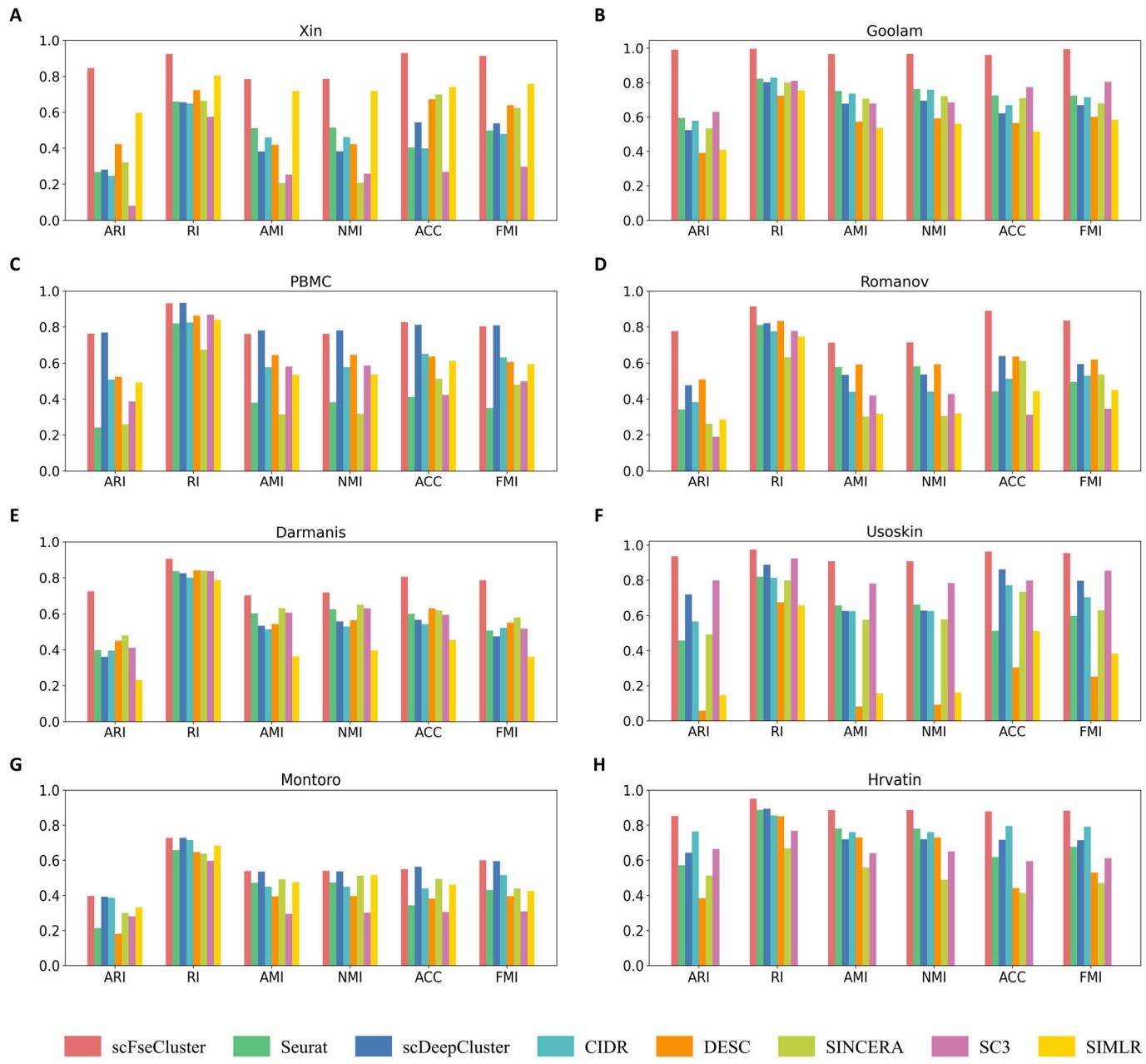

**Figure 3. Comparative analysis of clustering performance between scFseCluster and seven State-of-the-Art algorithms on eight single-cell RNA-seq datasets.**
Each method is evaluated with six metrics: ARI, RI, AMI, NMI, ACC, and FMI. Because of the huge sample size, an error is reported by SIMLR, therefore, the evaluation for SIMLR on dataset Hrvatin is not available. **(A)** Xin; **(B)** Goolam; **(C)** PBMC; **(D)** Romanov; **(E)** Darmanis; **(F)** Usoskin; **(G)** Montoro; **(H)** Hrvatin.

scRNA-seq dataset, shows that the performance of scFseCluster is comparable with that of scDeepCluster on all metrics (Fig 3C). Overall, scFseCluster is superior to seven existing methods for scRNA-seq clustering analysis.

## scFseCluster provides the best clustering visualization

To illustrate the effectiveness of dimension reduction, we applied t-SNE to visualize the final embedded points in two-dimensional (2D) space, which were learned by the FSQSSA in the scFseCluster

model. The 2D space representations of other methods are also plotted. Fig 4 shows that scFseCluster works well on all the datasets with different sample sizes, especially on the dataset Goolam. scFseCluster algorithm accurately separated all clusters with only two cells superimposed on each other. Overall, our method achieved a clustering accuracy of up to 85.05% for all datasets, significantly superior to other methods.

In addition, we found that both scDeepCluster and DESC are better for clustering visualization on datasets with moderate sample size (PBMC and Romanov). However, they perform poorly on other

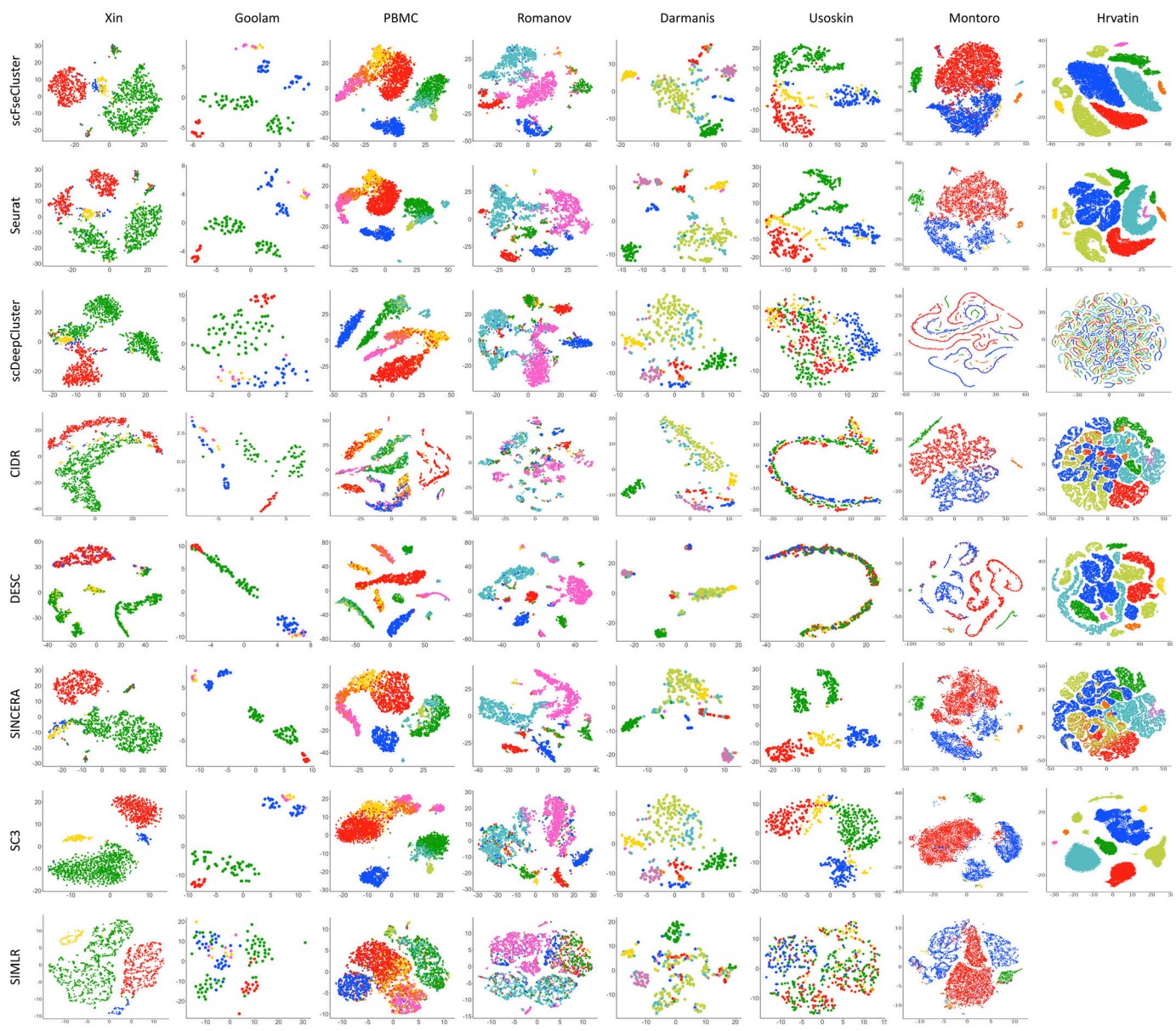

**Figure 4. Comparison of 2D visualization of embedded representations.**
Each point represents a cell. The distinct colors of the points represent the true labels. Because of the huge sample size, an error is reported by SIMLR; therefore, the evaluation for SIMLR on dataset Hrvatin is not available.

datasets. In summary, our scFseCluster model provides the best clustering visualization compared with seven other SOTA algorithms.

## Feature selection is an essential step in scRNA-seq clustering analysis

Unlike the experiments shown above, in this study, we applied the optimal gene sets generated by our model to seven other clustering methods to demonstrate if feature selection is an essential step in scRNA-seq clustering analysis. Fig 5 summarizes the default clustering method and the clustering performance with FSQSSA feature clustering performance (Supplemental Data 4). For most methods, the

optimal genes selected by FSQSSA provided better clustering performance. Across all datasets, ARI values were increase by an average of 0.11 and ACC by an average of 0.09 for the first six SOTA methods, indicating a significant improvement in the clustering performance of these methods. In some datasets, clustering performance of some methods improved significantly with the specified gene subsets. For example, Seurat's ARI value increased from 0.24 to 0.75 and the ACC value from 0.41 to 0.81 for clustering using the optimal gene set of FSQSSA features from the PBMC dataset. All of the above tests were well-controlled. The only difference between the blue and red bars is the feature selection procedure. Although these different clustering tools perform differently on different datasets, using features selected by FSQSSA can improve their

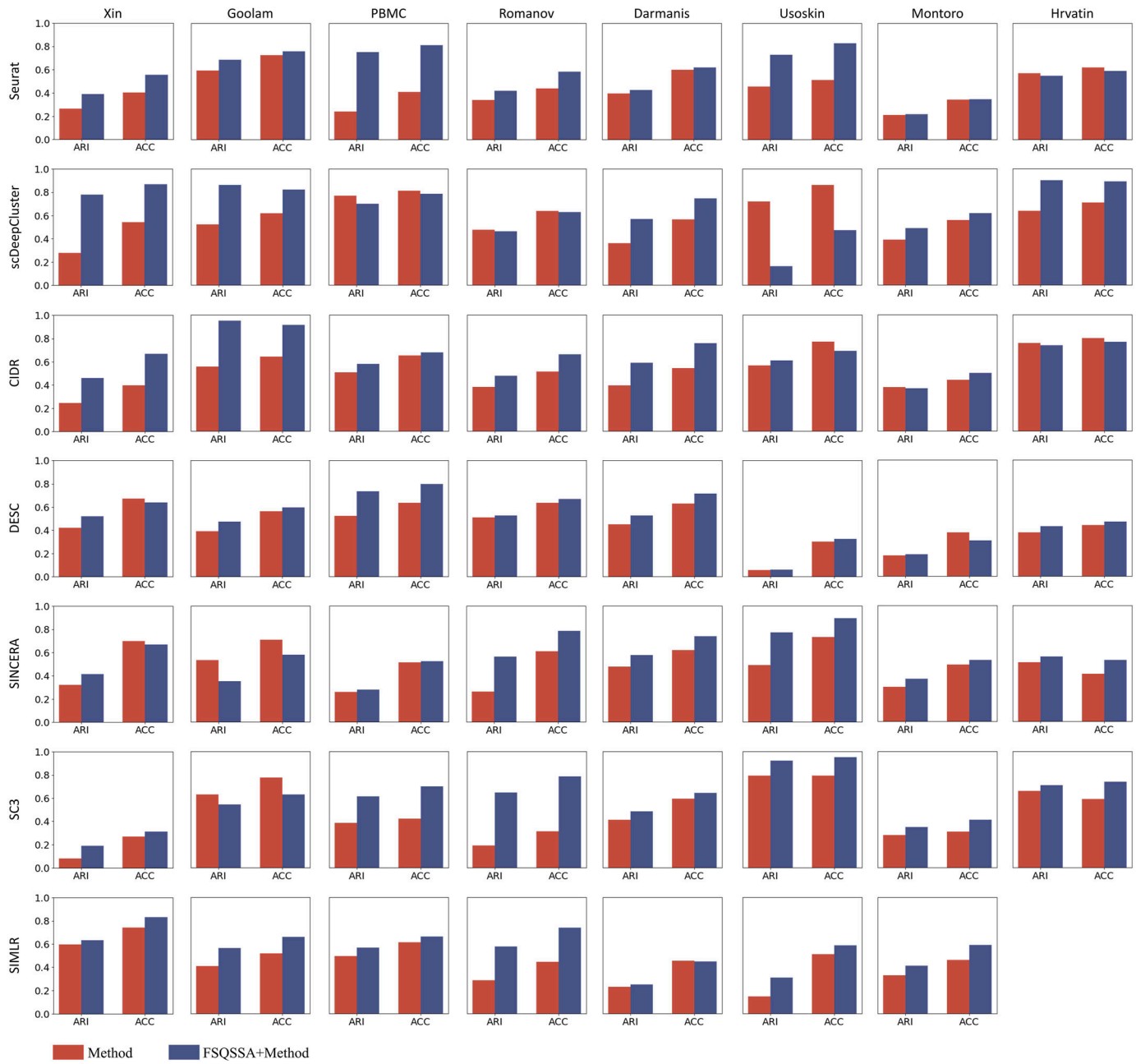

**Figure 5. Feature Selection based on Quantum Squirrel Search Algorithm significantly improves the accuracy of clustering.**
ARI and ACC are used to evaluate the performance of each State-of-the-Art algorithm with (blue) and without (red) feature selection processing. Because of the huge sample size, an error is reported by SIMLR; therefore, the evaluation for SIMLR on dataset Hrvatin is not available.

clustering accuracy. Taken together, we show the superior performance and broad applicability of FSQSSA, regardless of the clustering methods, experimental protocols, and dataset size.

### Computational cost is the main bottleneck of scFseCluster

Driven by a meta-heuristic method (FSQSSA), the computational cost of scFseCluster is valuable to be investigated. In this study, we

examined the running time of scFseCluster and other comparative algorithms on all the datasets. Table 2 suggests that the computational cost is mainly affected by the sample size. During the above methods, Seurat appears to be the most efficient algorithm for scRNA-seq clustering. Moreover, we separated all the datasets as two groups according to the sample size. Table 3 shows that our proposed model consumes an average of 119.14 s across four small datasets (cells < 2,000) and ranks fifth among all eight methods. However, the computational efficiency of scFseCluster is

**Table 2.  The running time of all the eight algorithms (unit: second).**

|  | scFseCluster | Seurat | scDeepCluster | CIDR | DESC | SINCERA | SC3 | SIMLR |
|---|---|---|---|---|---|---|---|---|
| Xin | 220.42 | 10.02 | 3,119.37 | 28.16 | 81.37 | 69.34 | 539.27 | 19.26 |
| Goolam | 41.62 | 10.82 | 260.85 | 0.99 | 21.05 | 1.80 | 47.16 | 1.18 |
| PBMC | 769.88 | 26.90 | 3,503.28 | 138.74 | 145.56 | 24.73 | 779.67 | 28.72 |
| Romanov | 313.09 | 12.48 | 481.85 | 64.50 | 90.35 | 156.59 | 263.36 | 25.88 |
| Darmanis | 123.37 | 2.15 | 440.47 | 2.69 | 313.09 | 2.22 | 41.58 | 2.19 |
| Usoskin | 91.15 | 4.17 | 696.90 | 5.39 | 123.37 | 7.26 | 67.57 | 120.71 |
| Montoro | 5,225.56 | 25.62 | 15,943.80 | 1,017.02 | 462.02 | 889.98 | 3,314.81 | 1,536.71 |
| Hrvatin | 17,103.09 | 107.68 | 26,331.62 | 2,174.59 | 1767.72 | 4,153.62 | 11,572.11 | NA |

**Table 3.  The average running time of all the eight algorithms on small-scale and large-scale datasets (unit: second).**

| Algorithm | Cells < 2,000 | Cells > 2,000 |
|---|---|---|
| Seurat | 6.79 | 43.17 |
| CIDR | 9.31 | 848.71 |
| SINCERA | 20.15 | 1,306.23 |
| SIMLR | 35.84 | NA |
| scFseCluster | 119.14 | 5,852.91 |
| DESC | 134.72 | 616.41 |
| SC3 | 173.90 | 3,982.49 |
| scDeepCluster | 1,129.40 | 11,565.14 |

dramatically decreased when the datasets include tens thousands of cells. In addition, we found that scDeepCluster performed the worst in this analysis, indicating that AE-based approaches appear to cause higher computational costs.

# Discussion

Clustering and cell-type identification are important steps in scRNA-seq data analysis. In this study, we proposed a novel computational framework scFseCluster for scRNA-seq data clustering analysis. Particularly, an excellent feature selection algorithm FSQSSA was encapsulated in the scFseCluster platform, which achieves optimal gene sets for clustering tasks. We tested scFseCluster on many scRNA-seq datasets from different species (mouse and human) and tissues (brain, pancreas, and embryo) and demonstrated that scFseCluster is capable of providing steady and precise cell clustering.

Compared with the other seven well-known SOTA methods, scFseCluster achieved higher clustering performance on datasets of various scales. As a wrapper strategy (58), FSQSSA implements feature selection through an enhanced Squirrel Search Algorithm, which reveals faster convergence, higher fitness value, and better robustness than other meta-heuristic methods. Further analysis demonstrated that integrating our FSQSSA module into various established methods can substantially improve their clustering performance. Moreover, we sum the frequency of each gene was

selected across all the 500 optimal solutions. After ranking, we selected the top 50 genes with the highest frequency. From Fig S2, we found that these genes were selected in more than half of the optimal solutions, indicating that they play the key role in distinguishing samples in different clusters. Moreover, we applied Seurat on each scRNA-seq dataset to identify cell-type markers. Seurat trends to determine one gene as marker for a predicted cluster of cells. And then, we checked if the markers provided by Seurat are overlapping with the top 50 genes. Table S1 shows that most of the markers are likely to be included in the optimal solution of FSQSSA. More details about this analysis can be found from the file 'SupplFiles' in our GitHub.

Overall, the major contributions of this work are as follows. First, our proposed feature selection model is more general because the wrapper-based model is easier to select the optimal subset of features. Moreover, the dimensionality of the data can be significantly reduced (around 852 genes) by using FSQSSA, which greatly saves the time overhead for downstream analysis. Finally, this study suggests that researchers should mainly focus on gene selection rather than clustering models when performing scRNA-seq clustering analysis.

Limitations still exist in the current study. Swarm intelligence algorithms are stochastic in nature, and multiple runs can lead to inconsistent final results. To guarantee strong robustness, each dataset in this study was run 500 times in parallel. Another disadvantage is that the number of clusters $K$ must be given in advance in scFseCluster. In practice, this parameter is usually unknown. Fortunately, we can implement spectral decomposition (59) or Louvain algorithm (60) to determine the optimal $K$. As to the high computational cost, we are planning to use Mojo (61) to reconstruct the code for speeding up the efficiency of scFseCluster. Currently, we are developing a light-weight algorithm with feature-ranking strategy, which may provide efficient and accurate gene selection on scRNA-seq clustering.

# Materials and Methods

### Data collection and preprocessing

We collected six publicly available scRNA-seq datasets containing cell-type annotations and gene expression values from

various scRNA-seq platforms, which can be downloaded from the Gene Expression Omnibus (62) and BioStudies (63). All the datasets are from different species, including mice and humans, and from different organs, such as the brain, pancreas, and embryo. The detailed information on the datasets is summarized in Table 1.

Assume that $X$ is a single cell reads count matrix, where $X_{ij}$ represents the count of $j-th$ gene of the $i-th$ cell. The same preprocessing process was followed for all datasets. First, the genes that have no counts in any cell will be filtered out. The expression matrix of the single-cell transcriptome was considered more suitable for clustering analysis (64). Therefore, we converted the reads count matrix into the expression matrix by normalization and $log2$ transform. The current gene expression matrix remains a high-dimensional sparse matrix, and those low-expressed genes have insufficient information to recognize the cell type. Accordingly, we screened the top $D$ HVGs by Scanpy package (65) (default $D = 2000$) and subsequently input the HVG matrix into the feature selection algorithm FSQSSA.

## Methodology of scFseCluster

The whole framework of scFseCluster includes three steps (Fig 6). Firstly, HVG selection is implemented on the normalized gene expression matrix. Secondly, FSQSSA is used to select the optimally selected genes based on the HVG expression matrix. Finally, the optimally selected gene expression matrix was input into the clustering module for cell type detection. If the number of clusters ($K$) is given, $K$-means algorithm will be called; otherwise, Louvain algorithm (66) will be started and the suitable value of $K$ will be estimated.

## FSQSSA for feature selection

In this section, we introduce the proposed feature selection algorithm FSQSSA (Fig S3), which is inspired by Squirrel Swarm Algorithm (52). Each feature is indicated by a "1" or "0," which respectively signifies that the feature is selected or unselected. In quantum-based optimization (67), each feature is represented by a quantum bit or Q-bit ($q$). Q-bit is the superposition of a "0" and "1," which is expressed in Dirac notation as $q = \alpha|0\rangle + \beta|1\rangle$ (68). The

values of $\alpha$ and $\beta$ correspond to the probability that the value of the Q-bit is "0" and "1," respectively. They must also obey the formula $|\alpha|^2 + |\beta|^2 = 1$. Because Q-bits are a linear superposition of probabilities, they are able to represent a more versatile population (69). Because the Q-bit uses the Dirac notation and cannot be directly involved in the operation, it is necessary to represent each feature using the angle $\theta$ of the Q-bit (70). The symbol $\theta$ is related to the probabilities $\alpha$ and $\beta$ as follows: $\theta = \tan^{-1}(\alpha/\beta)$, $\alpha = \cos\theta$, $\beta = \sin\theta$.

Each position of the flying squirrel represents an individual, which consists of $D$ Q-bits. Here, $D$ represents the total number of features. So, each individual ($Q_i$) can be represented by the following Equation (1):

$$Q_i = [q_{i1}, q_{i2}, ..., q_{iD}] = [\theta_{i1}, \theta_{i2}, ..., \theta_{iD}] \qquad (1)$$

The state of the $j-th$ element in $Q_i$ can be derived using Equation (2). $x_j^i$ is equal to 1 denotes the feature included in the feature subset; otherwise, it is not selected.

$$x_j^i = \begin{cases} 1, & if\ |\alpha|^2 \leq |\beta|^2 \\ 0, & otherwise \end{cases} \qquad (2)$$

The uniform distribution (Equation (3)) is used to assign the initial position of each flying squirrel.

$$Q_i = \theta_L + random(0,1) \times (\theta_U - \theta_L) \qquad (3)$$

where $\theta_L$ and $\theta_U$ are the lower and upper bounds of $i-th$ flying squirrel in $j-th$ dimension. In addition, $random(0,1)$ is a uniformly distributed random number in the range $[0,1]$.

The fitness function in the FSQSSA is an important metric for assessing the strength of individuals in a population. The fitness value reflects the goodness of fit of each candidate solution (optimal feature subset) to the objective problem. As a multi-objective problem, FSQSSA tries simultaneously minimizing the size of a subset of selected features and maximizing the clustering accuracy of a given subset of features. Based on the above basis, the fitness function constructed to achieve a balance between the two objectives for determining the solution, in this case, is defined as shown in Equation (4).

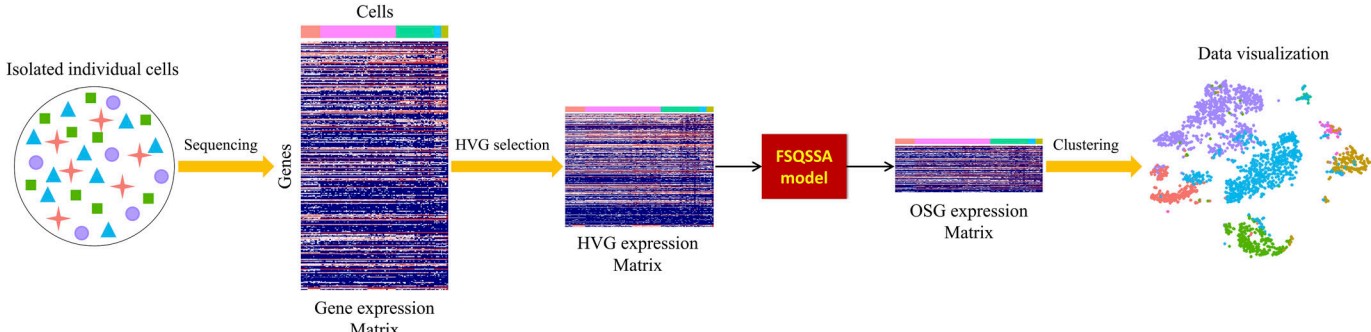

**Figure 6. Diagram of the proposed scFseCluster framework.**
HVG denotes a highly variable gene. OSG means optimally selected gene.

$$Fitness(S_i) = w \times SC(\hat{y}_i) + (1 - w) \times \left(1 - \frac{|S_i|}{D}\right) \qquad (4)$$

where $S_i$ represents the subset of features obtained by $i$-$th$ squirrel, and for each feature subset, this study uses the K-Means model for clustering, $\hat{y}_i$ means the clustering label of the output of the $i$-$th$ subset. The function $SC(\hat{y}_i)$ denotes the contour coefficient of the potential feature subset, and $|S_i|$ indicates the number of selected features. The parameter $w$ is a balance parameter that controls the clustering accuracy and feature selection rate. To ensure that our primary objective of maintaining accuracy is achieved, we set $w$ as 0.9 in our study (53, 71, 72, 73).

As mentioned earlier, three types of trees in the forest represent different food resource classes. To make FSQSSA achieve a better balance between exploration and exploitation, we assume that there are 50 trees in the forest; only 1 tree was top-ranked hickory, 3 second-ranked acorns, and 46 lowest-ranked normal trees. The number of squirrels matches the number of trees in the forest, with only one squirrel per tree.

Squirrels need to constantly search for more advanced resources in the forest to satisfy their requirements. The dynamic foraging process of a flying squirrel leads to three scenarios: (1) a squirrel flies from an acorn tree to a hickory tree; (2) a squirrel flies from a normal tree to an acorn tree; and (3) a squirrel flies from a normal tree directly to a hickory tree. It is hypothesized that in the absence of natural predators, a squirrel glides throughout the forest and effectively searches for food, whereas the presence of natural predators causes it to become alarmed and forced to flee to random locations. Natural enemies give each squirrel room to escape, which makes FSQSSA less likely to fall into a local optimum solution. We define the probability of the presence of a natural enemy as $P_{dp}$, which is equal to 0.1 by default. The squirrel's foraging process can be mathematically modeled as follows.

### Case 1

A squirrel flies from an acorn tree ($\theta_{at}$) to a hickory tree ($\theta_{ht}$). In this case, the new location of the squirrel can be obtained as follows:

$$\theta_{at}^{t+1} = \begin{cases} \theta_{at}^t + d_g \times G_c \times \left(\theta_{ht}^t - \theta_{at}^t\right), & R_1 \geq P_{dp} \\ Random\ location, & otherwise \end{cases} \qquad (5)$$

where $d_g$ is the random glide distance, defaulted between 0.3 and 0.7. $R_1$ is a random number ranging between $[0, 1]$. The $t$ denotes the current iteration. The balance between exploration and exploitation is achieved through the sliding constant $G_c$ in the equation, whose value significantly affects the algorithm's performance, which uses the default value of 1.9 in the standard Squirrel Search Algorithm.

### Case 2

Flying squirrel moves from a normal tree ($\theta_{nt}$) to an acorn tree. In this case, the new location of squirrels can be obtained as follows:

$$\theta_{nt}^{t+1} = \begin{cases} \theta_{nt}^t + d_g \times G_c \times \left(\theta_{at}^t - \theta_{nt}^t\right), & R_2 \geq P_{dp} \\ Random\ location, & otherwise \end{cases} \qquad (6)$$

where $R_2$ is a random number in the range $[0, 1]$.

### Case 3

Some of the squirrels in the normal tree fly directly to hickory trees. In this case, the new location of squirrels can be obtained as follows:

$$\theta_{nt}^{t+1} = \begin{cases} \theta_{nt}^t + d_g \times G_c \times \left(\theta_{ht}^t - \theta_{nt}^t\right), & R_3 \geq P_{dp} \\ Random\ location, & otherwise \end{cases} \qquad (7)$$

where $R_3$ is a random number in the range $[0, 1]$.

Seasonal changes can significantly affect the foraging activity of squirrels (74). They suffer substantial heat loss at low temperatures, and weather conditions force them to be less active in winter than fall (75). Squirrel movements are affected by changes in weather, hence the seasonal monitoring conditions retained in this study. The seasonal monitoring condition prevents FSQSSA from falling into a local optimum solution and enhances the exploratory ability of squirrels. The following steps are involved in modeling the behavior.

(a) First, calculate the seasonal constant ($S_c$) using Equation (8).

$$S_c^t = \sqrt{\sum_{k=1}^{d} \left(\theta_{at,k}^t - \theta_{ht,k}^t\right)^2} \qquad (8)$$

Where $t = 1, 2, 3$.

(b) Check the seasonal monitoring condition, that is, $S_c^t < S_{min}$ where $S_{min}$ is the minimum value of seasonal constant computed as:

$$S_{min} = \frac{10E^{-6}}{(365)^{t/(t_m/2.5)}} \qquad (9)$$

Where $t$ and $t_m$ are the current and maximum iteration values, respectively. The value $S_{min}$ affects the exploration and exploitation capabilities of the proposed method. Larger value of $S_{min}$ promotes exploration, whereas smaller values of $S_{min}$ enhance the exploitation capability of the algorithm. For any effective metaheuristic, there must be a proper balance between these two phases (76).

(c) If seasonal monitoring condition is found to be true, those flying squirrels unable to explore the forest for optimal winter food sources will be randomly relocated. Because the normal trees have the lowest level of food resources, FSQSSA assumes that only squirrels in the normal trees will be forced to migrate randomly in search of better food sources. The random migration of squirrels is given by Equation (10).

$$\theta_{nt}^{new} = \theta_L + L\acute{e}vy(n) \times (\theta_U - \theta_L) \tag{10}$$

Lévy flight is a powerful mathematical tool used for improving global exploration capabilities of various metaheuristic algorithms (77). *Lévy* flight helps to find new candidate solutions far from the current best solution.

### Performance metrics

We aggregate six quality metrics (78) including Rand Index (*RI*), Adjusted Rand Index (*ARI*), Normalized Mutual Information (*NMI*), Adjusted Mutual Information (*AMI*), Accuracy (*ACC*), and Fowlkes–Mallows Index (*FMI*) to access the clustering performance of the scFseCluster model. These metrics are defined as follows:

$$RI = \frac{Number\ of\ pair-wise\ correct\ predictions}{Total\ number\ of\ possible\ pairs} \tag{11}$$

$$ARI = \frac{Number\ of\ pair-wise\ true\ positive\ prediction - E[RI]}{Average\ number\ of\ pairs\ in\ same\ cluster\ for\ actual\ and\ predicted - E[RI]} \tag{12}$$

$$NMI = \frac{MI}{[H(U) + H(V)]/2} \tag{13}$$

$$AMI = \frac{MI - E[MI]}{[H(U) + H(V)]/2 - E[MI]} \tag{14}$$

$$ACC = \frac{TP + TN}{TP + FP + TN + FN} \tag{15}$$

$$FMI = \frac{TP}{\sqrt{(TP + FP) \times (TP + FN)}} \tag{16}$$

*RI* and *ARI* measure the similarity between the cluster assignments by making pair-wise comparisons (79). *NMI* and *AMI* measure the agreement between the cluster assignments (53). $H(U)$ and $H(V)$ denote the entropy of actual and predicted cluster assignments, respectively. *NM* is equal to $\sum_{i=1}^{|U|}\sum_{j=1}^{|V|}P(i,j)\log[P(i,j)/P(i)P'(j)]$. $P(i)$ and $P'(j)$ represent the probability of data occurring in Cluster *i* (actual) and Cluster *j* (predicted). *FMI* measures the correctness of the cluster assignments using pairwise precision and recall (53). The definition for *TP*, *TN*, *FP*, and *FN* is done by counting the number of pairwise samples if they are allocated in the same or different cluster for the predicted and actual labels.

### Comparison analysis

To prove the effectiveness, we carried out a comparison analysis from two aspects. In one aspect, we compared FSQSSA with metaheuristic methods, including the standard Squirrel Search Algorithm (Squirrel) (52), Enhanced Salp Swarm Algorithm (Salp) (53), ABC (54), and Genetic Algorithm (GA) (55). All the comparative algorithms share the same fitness function (Equation (4)). In addition,

all of them were iterated 100 times, with 50 individuals in each iteration of the population. In particular, it should be noted that the standard Squirrel and ABC can only solve continuous optimization problems, whereas feature selection is typically a discrete optimization problem. For this purpose, we apply the sigmoid function to these two algorithms to obtain the feature subset.

In another aspect, we compared our scFseCluster algorithm with seven SOTA methods for scRNA-seq data clustering, which includes two deep learning approaches scDeepCluster (35) and DESC (33), and five machine learning methods: Seurat (28), CIDR (56), SINCERA (13), SC3 (29), and SIMLR (57). We respect all the steps of other methods without any additional extraneous operations. Table S2 summarizes the details of these methods.

### Simulation environment

The codes of scFseCluster were developed and debugged by using Tensorflow 2.7.0 and Python 3.7.0 under the environment with GPU 3090 and 32G RAM. All the simulations and parallel calculations (500 repeats) were performed on NJAU-HPC with 16G memory of graphics card.

## Data Availability

The source code of scFseCluster framework is available at GitHub: https://github.com/wzqwtt/scFseClusterV1.0. All the processed data are placed in the link: http://cdsic.njau.edu.cn/data/scFseClusterV1.0. The researchers can also download the data and source code from FigShare: https://figshare.com/s/a0c082d8244c942b5515.

## Supplementary Information

## Acknowledgements

The authors thank the Bioinformatics Center, Nanjing Agricultural University for the High-Performance Computing platform (NJAU-HPC). This work was supported by the Fundamental Research Funds for the Central Universities (No. KYCXJC2023001, No. KYCXJC2022005), Natural Science Foundation of Jiangsu Province (No. BK20211210), and Agricultural Science and Technology Innovation Foundation of Jiangsu Province (No. CX (23) 3125). This work was partially supported by the startup award of new professors at Nanjing Agricultural University (No. 106/804001).

### Author Contributions

Z Wang: formal analysis and methodology.
X Xie: resources and software.
S Liu: validation.

Z Ji: conceptualization, funding acquisition, project administration, and writing—review and editing.

## Conflict of Interest Statement

The authors declare that they have no conflict of interest.

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
