## [Reviewer comments · Life Science Alliance]

Life Science Alliance

scFseCluster: A Feature Selection Enhanced Clustering for Single Cell RNA-seq Data

Zongqin Wang, Xiaojun Xie, Shouyang Liu, and Zhiwei Ji

DOI: <https://doi.org/10.26508/lsa.202302103>

Corresponding author(s): Zhiwei Ji, Nanjing Agricultural University

Review Timeline:

Submission Date:	2023-04-20
Editorial Decision:	2023-05-22
Revision Received:	2023-08-25
Editorial Decision:	2023-09-19
Revision Received:	2023-09-21
Accepted:	2023-09-22

Transaction Report:

May 22, 2023

Re: Life Science Alliance manuscript #LSA-2023-02103-T

Zhiwei Ji
Nanjing Agricultural University, College of Artificial Intelligence

Dear Dr. Ji,

Thank you for submitting your manuscript entitled "scFseCluster: A Feature Selection Enhanced Clustering for Single Cell RNA-seq Data" to Life Science Alliance. The manuscript was assessed by expert reviewers, whose comments are appended to this letter. We invite you to submit a revised manuscript addressing the Reviewer comments.

Thank you for this interesting contribution to Life Science Alliance. We are looking forward to receiving your revised manuscript.

Sincerely,

B. MANUSCRIPT ORGANIZATION AND FORMATTING:

Reviewer #1 (Comments to the Authors (Required)):

In this study, the authors proposed a hybrid computational framework scFseCluster for single cell RNA-seq clustering analysis, which can be used for cell type identification. scFseCluster is driven by a metaheuristic algorithm FSQSSA and implements cell clustering based on predicted gene sets. Overall, this work revealed a great significance and the manuscript is well-written. The simulation results proved that their tool works very well. However, several concerns should be carefully addressed:

- 1) It will be better if the authors improve the summary of the existing methods in the introduction part. Particularly, some typical algorithms (e.g., DESC, scDeepCluster) mentioned by the authors were established in 2019-2020. How about the development in this area from 2020-2022?
- 2) In Figure 3, please analyze why the variation on some datasets are large, e.g., Romanov.
- 3) The superiority of the proposed framework should be clearly clarified in the theoretical and practical perspective.
- 4) Moreover, I suggest the authors discuss the reasons why the computational cost of scFseCluster is high? And is there any possible strategy to speed up this tool in the near future?
- 5) In the Github, a Jupyter notebook (ipynb format) of the code should be included with comments so that the process of each step can be checked more precisely.
- 6) The writing still needs to be further improved by a native speaker.

Reviewer #2 (Comments to the Authors (Required)):

In this study, the authors have presented scFseCluster, a novel computational framework for scRNA-seq clustering analysis. By incorporating the FSQSSA metaheuristic algorithm, scFseCluster extracts an optimal gene set, ensuring superior cell clustering performance. Through simulation experiments and comparisons with seven state-of-the-art algorithms, scFseCluster demonstrated exceptional performance on six benchmark scRNA-seq datasets. Moreover, their analysis highlighted the significant benefits of feature selection on high-variable genes (HVGs) for improving clustering accuracy. Overall the paper is well written and presented. Results are clearly presented and interpreted but I have some concerns that needs to be addressed. Below are my major concerns:

- Current single-cell clustering and visualization methods often reduce dimensionality by selecting the most variable features among cells. However, it's unclear whether these selected features are effective in distinguishing between cell types and states, as discriminative features may not overlap with the most variable features, particularly in tumors with dosage effects and driver variants. Is it possible to apply the metaheuristic algorithm FSQSSA to all genes, rather than just highly variable genes? Are the genes selected from all genes similar to those chosen from highly variable genes?
- Are the genes selected using FSQSSA overlapping with the well-known cell type markers?
- The paper lacks a clear explanation of the six metrics (ARI, RI, AMI, NMI, ACC, and FMI) used in the comparisons. The formula and definition of these metrics are missing, which is essential for readers to understand and interpret the results accurately. I recommend that the authors provide a detailed description of each metric, including their formulas and definitions, in the methodology section.
- In the method section the authors stated that "To ensure that our primary objective of maintaining accuracy is achieved, we set w as 0.9 in our study." However, it is unclear how this parameter affects the results and what specific role it plays in the proposed method. I recommend that the authors provide a more detailed explanation of the parameter " w " and its impact on the results obtained.
- The method details of FSQSSA are somewhat challenging to follow. To address this concern, I recommend that the authors provide a figure illustrating the method details using a toy dataset. This visual representation will significantly aid in understanding the step-by-step process of FSQSSA and how it extracts the optimal gene set. By providing a clear visual guide, readers will have a better grasp of the methodology and its implementation.

Reviewer #3 (Comments to the Authors (Required)):

Wang et al. present a new clustering method for single cell RNA sequencing data, scFseCluster, which aims to enhance and accelerate clustering with a new approach to feature selection. FSQSSA metaheuristics is used to extract an optimal gene set, bypassing dimension reduction steps such as PCA, for maximum preservation of information. The authors demonstrate good performance of scFseCluster versus other commonly used tools in the field, against six benchmarking datasets. The tool can be used to analyze data all the way to 2-d visualizations, or used as drop-in replacement for feature selection in other workflows.

Major comments:

1. As the cost of scRNA-seq decreases, the scale of data generated has increased. However, the largest dataset benchmarked here is only 4271 cells. Performance and run-time reporting on applying scFseCluster to much larger datasets, on the scale of tens or hundreds of thousands of cells, would further support the utility of the method.
2. Run time and resource usage of scFseCluster seems important enough to warrant including the main text/tables.

Minor comments:

1. SOTA and FSQSSA acronyms appear but are never explained in the abstract or text.
2. For Fig1, is 2-d visualization via tSNE or UMAP worth adding as an additional step (or addition of text) in the diagram, since the authors comment on how well scFseCluster works on that step?
3. Please clarify, is FSQSSA feature selection results deterministic, or does a seed need to be set to ensure the exact same results every time the algorithm is run?

Response to reviewers

Title: scFseCluster: A Feature Selection Enhanced Clustering for Single Cell RNA-seq Data

Manuscript number: LSA-2023-02103-T. R1

Authors: Zongqin Wang, Xiaojun Xie, Shouyang Liu, Zhiwei Ji*

Reviewer's comments to Authors:

Reviewer #1 (Comments to the Authors (Required)):

In this study, the authors proposed a hybrid computational framework scFseCluster for single cell RNA-seq clustering analysis, which can be used for cell type identification. scFseCluster is driven by a metaheuristic algorithm FSQSSA and implements cell clustering based on predicted gene sets. Overall, this work revealed a great significance and the manuscript is well-written. The simulation results proved that their tool works very well. However, several concerns should be carefully addressed:

1) It will be better if the authors improve the summary of the existing methods in the introduction part. Particularly, some typical algorithms (e.g., DESC, scDeepCluster) mentioned by the authors were established in 2019-2020. How about the development in this area from 2020-2022?

Thanks for pointing this out.

According to the literature survey, we found that there are three representative models (scGNN [1], scCAEs [2], scDHA [3]) can be added to the introduction part, which were proposed in the last 2 years. Therefore, we improved the summary of the existing methods. Please refer the revised manuscript.

2) In Figure 3, please analyze why the variation on some datasets are large, e.g., Romanov.

We thank the reviewer's comments.

Yes. As the reviewer mentions, obvious variations may exist on some datasets. This is essentially caused by the randomness of the metaheuristic search algorithms. An excellent swarm intelligence algorithm (SIA) is expected to provide a stable solution under multiple independent repeats [4]. However, SIAs will face great challenge when the dimensionality of data is high.

*In this revision, we added two scRNA-seq datasets with larger sample size and updated the experimental results. Our results show that the variation induced by FSQSSA on most of datasets is very small. Please refer the revised **Figure 3**.*

3) The superiority of the proposed framework should be clearly clarified in the theoretical and practical perspective.

We thank the reviewer's suggestion.

In the section of Discussion, we improved the advantage of our proposed method. Please check the revised manuscript.

4) Moreover, I suggest the authors discuss the reasons why the computational cost of scFseCluster is high? And is there any possible strategy to speed up this tool in the near future?

We thank the reviewer's comments.

Due to the high dimensionality of scRNA-seq data, selecting optimal gene set is a NP-hard problem [5, 6]. In other word, the search space of FSQSSA is very large. Therefore, FSQSSA algorithm implement multiple iterations to approximate the global solution. In addition, we need to repeat the FSQSSA over 500 repeats via parallel computing to reduce the variation of model outcome as the feature selection result is not deterministic. Overall, the high computational cost is the main limitation for metaheuristic algorithms.

In the near future, we may use Mojo to rewrite the code to speed up the efficiency of scFseCluster. As a revolutionary programming language, Mojo is 35,000 times faster than Python [7]. Furthermore, Mojo will leverage the entire ecosystem of Python libraries built on a brand-new codebase. The fragmentation in the ecosystem and deployment challenges faced by the Python community is resolved by Mojo.

5) In the Github, a Jupyter notebook (ipynb format) of the code should be included with comments so that the process of each step can be checked more precisely.

We thank the reviewer's comments.

A Jupyter notebook of the code was included in the Github. Please refer the link: <https://github.com/wzqwt/scFseClusterV1.0>.

6) The writing still needs to be further improved by a native speaker.

We thank the reviewer's comments.

The writing has been improved. Language and technical errors were corrected. Please check the revised manuscript.

Reviewer #2 (Comments to the Authors (Required)):

In this study, the authors have presented scFseCluster, a novel computational framework for scRNA-seq clustering analysis. By incorporating the FSQSSA metaheuristic algorithm, scFseCluster extracts an optimal gene set, ensuring superior cell clustering performance. Through simulation experiments and comparisons with seven state-of-the-art algorithms, scFseCluster demonstrated exceptional performance on six benchmark scRNA-seq datasets. Moreover, their analysis

highlighted the significant benefits of feature selection on high-variable genes (HVGs) for improving clustering accuracy. Overall the paper is well written and presented. Results are clearly presented and interpreted but I have some concerns that needs to be addressed. Below are my major concerns:

- Current single-cell clustering and visualization methods often reduce dimensionality by selecting the most variable features among cells. However, it's unclear whether these selected features are effective in distinguishing between cell types and states, as discriminative features may not overlap with the most variable features, particularly in tumors with dosage effects and driver variants. Is it possible to apply the metaheuristic algorithm FSQSSA to all genes, rather than just highly variable genes? Are the genes selected from all genes similar to those chosen from highly variable genes?

We thank the reviewer's comments.

- 1) *In the theoretical perspective, the metaheuristic algorithm FSQSSA can be directly applied to all genes. However, this is not recommended. Firstly, the dimensionality of the original scRNA-seq data is very high, which will cause high computational cost of FSQSSA. Secondly, the genes with missing values or low expressions should be removed before data analysis because they provide little power in distinguishing between cell types. Our study starts from highly variable genes (HVGs) and identifies an optimal gene set to significantly improve the clustering performance.*
- 2) *As the same as other wrapper models [8, 9], FSQSSA algorithm uses heuristic search strategy to obtain the optimal feature subset. In other word, the optimization process involves searching a set of potential feature subset and evaluating their goodness. Hence, the genes selected from all genes are not the same as that chosen from HVGs.*

- Are the genes selected using FSQSSA are overlapping with the well-known cell type markers?

We thank the reviewer's question.

The genes selected using FSQSSA is a subset of HVGs. In our study, the default numbers of HVGs for model input is 2000.

*In this revision, we sum the frequency of each gene was selected across all the 500 optimal solutions. After ranking, we selected top 50 genes with the highest frequency. From **Figure S2**, we found that these genes were selected in more than half of the optimal solutions, indicating that they play the key role in distinguishing samples in different clusters.*

*Moreover, we applied Seurat [10] on each scRNA-seq dataset to identify cell type markers. Seurat trends to determine one gene as marker for a predicted cluster of cells. And then, we checked if the markers provided by Seurat are overlapping with the top 50 genes. **Table S2** shows that most of markers are likely to be included in the optimal solution of FSQSSA.*

- The paper lacks a clear explanation of the six metrics (ARI, RI, AMI, NMI, ACC, and FMI) used in the comparisons. The formula and definition of these metrics are missing, which is essential for

readers to understand and interpret the results accurately. I recommend that the authors provide a detailed description of each metric, including their formulas and definitions, in the methodology section.

We thank the reviewer's suggestions.

The mathematical formulas for all the metrics were added. Please check the revised manuscript.

- In the method section the authors stated that "To ensure that our primary objective of maintaining accuracy is achieved, we set w as 0.9 in our study." However, it is unclear how this parameter affects the results and what specific role it plays in the proposed method. I recommend that the authors provide a more detailed explanation of the parameter " w " and its impact on the results obtained.

Thanks for pointing this out.

In our study, the fitness function (Eq. (4)) maximizes two factors: one is the clustering accuracy of a selected feature subset; the other is the number of genes filtered out. The coefficient w in Eq. (4) is defined to weight the effect of each factor on fitness. To guarantee the clustering accuracy as the primary factor, the value of w is set to 0.9. Similar strategies to determine the value of w were reported in the previous works [4, 11-13].

- The method details of FSQSSA are somewhat challenging to follow. To address this concern, I recommend that the authors provide a figure illustrating the method details using a toy dataset. This visual representation will significantly aid in understanding the step-by-step process of FSQSSA and how it extracts the optimal gene set. By providing a clear visual guide, readers will have a better grasp of the methodology and its implementation.

We thank the reviewer's suggestion.

*In this revision, we generate a flow chart of the FSQSSA algorithm and its operations as it is applied to gene selection. Please refer the **Figure S3**.*

Reviewer #3 (Comments to the Authors (Required)):

Wang et al. present a new clustering method for single cell RNA sequencing data, scFseCluster, which aims to enhance and accelerate clustering with a new approach to feature selection. FSQSSA metaheuristics is used to extract an optimal gene set, bypassing dimension reduction steps such as PCA, for maximum preservation of information. The authors demonstrate good performance of scFseCluster versus other commonly used tools in the field, against six benchmarking datasets. The tool can be used to analyze data all the way to 2-d visualizations, or used as drop-in replacement for feature selection in other workflows.

Major comments:

1. As the cost of scRNA-seq decreases, the scale of data generated has increased. However, the largest dataset benchmarked here is only 4271 cells. Performance and run-time reporting on applying scFseCluster to much larger datasets, on the scale of tens or hundreds of thousands of cells, would further support the utility of the method.

We thank the reviewer's suggestions.

*Yes, it is necessary to test scFseCluster on much larger datasets. In this revision, we added two new scRNA-seq datasets for further analysis: 1) The dataset GSE103354 was generated by Montoro's group by sequencing 7193 mouse tracheal epithelial cells [14]; 2) The dataset GSE102827 was released by Hrvatin et al. They profiled 48266 cells from mouse visual cortex [15]. The simulation experiments show that our method also works well on these two new datasets. Thus, we conclude that our method is applicable to datasets with various scales. We updated the Results part and modified the **Figure 2-6**.*

Please refer the revised manuscript.

2. Run time and resource usage of scFseCluster seems important enough to warrant including the main text/tables.

We thank the reviewer's suggestion.

*The **Table S2** in the original **Supplementary materials** was moved to the main text. Please refer the new **Table 2-3** in the revised manuscript.*

Minor comments:

1. SOTA and FSQSSA acronyms appear but are never explained in the abstract or text.

Thanks for pointing this out.

We highlighted the full names of SOTA and FSQSSA in the Abstract.

2. For Fig1, is 2-d visualization via tSNE or UMAP worth adding as an additional step (or addition of text) in the diagram, since the authors comment on how well scFseCluster works on that step?

Thanks for pointing this out.

Figure 1 has been modified. Please check the revised manuscript.

3. Please clarify, is FSQSSA feature selection results deterministic, or does a seed need to be set to ensure the exact same results every time the algorithm is run?

We thank the reviewer's comments.

As the same as other metaheuristic algorithms (e.g., GA [16], PSO [17, 18], ABC [19, 20], et al.), the result of FSQSSA algorithm is also not deterministic. The initialization of population is the main reason for the randomness of the optimal solution. In addition, the evolutionary operations performed by these algorithms during the iterative search is also one of the reasons for the randomness. Therefore, we clarify this question as follows:

-
- 1) FSQSSA algorithm starts from a random population, which corresponds to a set of initial solutions. It quickly converges to an optimal solution after a series of iteration.
 - 2) FSQSSA can't guarantee to obtain the exact same result after independent repeats.
 - 3) **Figure 3** shows that the variation of all the optimal solutions returned by FSQSSA is very small on the most of datasets.

In addition, we sum the frequency of each gene was selected across all the 500 optimal solutions of FSQSSA after parallel computing. After ranking, we selected top 50 genes with the highest frequency. From **Figure S2**, we found that these genes were selected in more than half of the optimal solutions, indicating that they play the key role in distinguishing samples in different clusters.

Moreover, we applied Seurat [10] on each scRNA-seq dataset to identify cell type markers. Seurat tends to determine one gene as marker for a predicted cluster of cells. And then, we checked if these markers provided by Seurat are overlapping with the top 50 genes. **Table S2** shows that most of markers are likely to be included in the optimal solution of FSQSSA.

References

1. Wang J, Ma A, Chang Y, Gong J, Jiang Y, Qi R, Wang C, Fu H, Ma Q, Xu D: **scGNN is a novel graph neural network framework for single-cell RNA-Seq analyses.** *Nat Commun* 2021, **12**(1):1882.
2. Hu H, Li Z, Li X, Yu M, Pan X: **ScCAEs: deep clustering of single-cell RNA-seq via convolutional autoencoder embedding and soft K-means.** *Brief Bioinform* 2022, **23**(1).
3. Tran D, Nguyen H, Tran B, La Vecchia C, Luu HN, Nguyen T: **Fast and precise single-cell data analysis using a hierarchical autoencoder.** *Nat Commun* 2021, **12**(1):1029.
4. Xie X, Xia F, Wu Y, Liu S, Yan K, Xu H, Ji Z: **A Novel Feature Selection Strategy Based on Salp Swarm Algorithm for Plant Disease Detection.** *Plant Phenomics* 2023, **5**:0039.
5. Naghibi T, Hoffmann S, Pfister B: **Convex approximation of the NP-hard search**

-
- problem in feature subset selection.** *2013 IEEE International Conference on Acoustics, Speech and Signal Processing* 2013:3273-3277.
6. Brezočnik L, Fister I, Podgorelec V: **Swarm Intelligence Algorithms for Feature Selection: A Review.** *Applied sciences* 2018:1-31.
 7. Zawdie K, Drob D, Huba J, Coker C: **Effect of time-dependent 3-D electron density gradients on high angle of incidence HF radiowave propagation.** *Radio Science* 2016, **51(7)**:1131-1141.
 8. Ruiz R, Riquelme J, Aguilar-Ruiz J: **Heuristic Search over a Ranking for Feature Selection.** *International Work-Conference on Artificial Neural Networks* 2005:742-749.
 9. Kaur A, Guleria K, Trivedi N: **Feature Selection in Machine Learning: Methods and Comparison.** *2021 International Conference on Advance Computing and Innovative Technologies in Engineering (ICACITE)* 2021:789-795.
 10. Satija R, Farrell JA, Gennert D, Schier AF, Regev A: **Spatial reconstruction of single-cell gene expression data.** *Nat Biotechnol* 2015, **33(5)**:495-502.
 11. Asif M, Nagra A, Ahmad M, Masood K: **Feature Selection Empowered by Self-Inertia Weight Adaptive Particle Swarm Optimization for Text Classification.** *Applied Artificial Intelligence* 2020, **36(1)**:282-297.
 12. Xue B, Zhang M, Browne W: **Particle Swarm Optimization for Feature Selection in Classification: A Multi-Objective Approach.** *IEEE Transactions on Cybernetics* 2013, **43(6)**:1656-1671.
 13. Xue Y, Xue B, Zhang M: **Self-adaptive particle swarm optimization for large-scale feature selection in classification.** *ACM Transactions on Knowledge Discovery from*

Data 2019, **13**(5):1-27.

14. Montoro DT, Haber AL, Biton M, Vinarsky V, Lin B, Birket SE, Yuan F, Chen S, Leung HM, Villoria J *et al*: **A revised airway epithelial hierarchy includes CFTR-expressing ionocytes.** *Nature* 2018, **560**(7718):319-324.
15. Hrvatin S, Hochbaum DR, Nagy MA, Cicconet M, Robertson K, Cheadle L, Zilionis R, Ratner A, Borges-Monroy R, Klein AM *et al*: **Single-cell analysis of experience-dependent transcriptomic states in the mouse visual cortex.** *Nat Neurosci* 2018, **21**(1):120-129.
16. Singh V, Misrab A: **Detection of plant leaf diseases using image segmentation and soft computing techniques.** *Information Processing in Agriculture* 2017, **4**(1):41-49.
17. Ji Z, Wang B: **Identifying potential clinical syndromes of hepatocellular carcinoma using PSO-based hierarchical feature selection algorithm.** *BioMed Research International* 2014, **2014**:1-12.
18. Song X, Zhang Y, Guo Y, Sun X, Wang Y: **Variable-Size Cooperative Coevolutionary Particle Swarm Optimization for Feature Selection on High-Dimensional Data.** *IEEE Transactions on Evolutionary Computation* 2020, **24**(5):882-895.
19. Mohammadi F, SanieeAbadeh M: **Image steganalysis using a bee colony based feature selection algorithm.** *Engineering Applications of Artificial Intelligence* 2014, **31**:35-43.
20. Agrawal V, Chandra S: **Feature selection using Artificial Bee Colony algorithm for medical image classification.** *2015 Eighth International Conference on Contemporary Computing (IC3)* 2015:1-6.

September 19, 2023

RE: Life Science Alliance Manuscript #LSA-2023-02103-TR

Prof. Zhiwei Ji
Nanjing Agricultural University
Computer Science
No. 1 Weigang Rd.
Nanjing, Jiangsu 210095
China

Dear Dr. Ji,

Thank you for submitting your revised manuscript entitled "scFseCluster: A Feature Selection Enhanced Clustering for Single Cell RNA-seq Data". We would be happy to publish your paper in Life Science Alliance pending final revisions necessary to meet our formatting guidelines.

- please upload all figure files as individual ones, including the supplementary figure files; all figure legends should only appear in the main manuscript file
- please add the Twitter handle of your host institute/organization as well as your own or/and one of the authors in our system
- please add your main, supplementary figure, and table legends to the main manuscript text after the references section
- please exclude figures from the manuscript text and upload them separately
- we encourage you to revise the figure legends for figures 4 and S1 such that the figure panels are introduced in an alphabetical order
- please add callouts for Figures 3A-H; 4A, G; S1A-H to your main manuscript text

A. FINAL FILES:

B. MANUSCRIPT ORGANIZATION AND FORMATTING:

Sincerely,

Reviewer #2 (Comments to the Authors (Required)):

There is a substantial improvement in the paper. Authors have made the requested changes. I believe the current version is suitable for publication.

September 22, 2023

RE: Life Science Alliance Manuscript #LSA-2023-02103-TRR

Prof. Zhiwei Ji
Nanjing Agricultural University
Computer Science
No. 1 Weigang Rd.
Nanjing, Jiangsu 210095
China

Dear Dr. Ji,

Thank you for submitting your Methods entitled "scFseCluster: A Feature Selection Enhanced Clustering for Single Cell RNA-seq Data". It is a pleasure to let you know that your manuscript is now accepted for publication in Life Science Alliance. Congratulations on this interesting work.

DISTRIBUTION OF MATERIALS:

Again, congratulations on a very nice paper. I hope you found the review process to be constructive and are pleased with how the manuscript was handled editorially. We look forward to future exciting submissions from your lab.

Sincerely,
